# Observed multi-decadal trends in subsurface temperature adjacent to the East Australian Current

**Michael P. Hemming**[1]**, Moninya Roughan**[1]**, Neil Malan**[1]**, and Amandine Schaeffer**[1,2]

[1]Coastal and Regional Oceanography Lab, School of Biological, Earth & Environmental Sciences, CE1 University of New South Wales, Sydney, New South Wales, Australia
[2]School of Mathematics and Statistics, University of New South Wales, Sydney, New South Wales, Australia

**Correspondence:** Michael P. Hemming (m.hemming@unsw.edu.au)

**Abstract.** Sea surface temperature observations have shown that western boundary currents, such as the East Australian Current (EAC), are warming faster than the global average. However, we know little about coastal temperature trends inshore of these rapidly warming regions, particularly below the surface. In addition to this, warming rates are typically estimated linearly, making it difficult to know how these rates have changed over time. Here we use long-term in situ temperature observations through the water column at five coastal sites between approximately 27.3–42.6° S to estimate warming trends between the ocean surface and the bottom. Using an advanced trend detection method, we find accelerating warming trends at multiple depths in the EAC extension region at 34.1 and 42.6° S. We see accelerating trends at the surface and bottom at 34.1° S but similar trends in the top 20 m at 42.6° S. We compare several methods, estimate uncertainty, and place our results in the context of previously reported trends, highlighting that magnitudes are depth-dependent, vary across latitude, and are sensitive to the data time period chosen. The spatial and temporal variability in the long-term temperature trends highlight the important role of regional dynamics against a background of broad-scale ocean warming. Moreover, considering that recent studies of ocean warming typically focus on surface data only, our results show the necessity of subsurface data for the improved understanding of regional climate change impacts.

## 1 Introduction

Globally averaged surface air temperatures have increased by approximately 1.3 °C since the start of the industrial revolution (Hartmann et al., 2013; Masson-Delmotte et al., 2021), and more than 90 % of the excess heat has been absorbed by the oceans since the 1950s (Levitus et al., 2012). Surface ocean temperatures in western boundary current regions have warmed 2 to 3 times the global rate since the 1990s (Wu et al., 2012).

The East Australian Current (EAC), the western boundary current of the South Pacific subtropical gyre, transports heat poleward (Archer et al., 2017). It typically separates at 30 to 32.5° S (Cetina-Heredia et al., 2014) and extends eastward towards New Zealand (Godfrey et al., 1980; Oke et al., 2019), while at the same time it produces mesoscale warm-core eddies (Nilsson and Cresswell, 1980).

The EAC has previously been reported as strengthening (Cai et al., 2005; Roemmich et al., 2007) and penetrating further south (Hill et al., 2008; Ridgway, 2007; Cetina-Heredia et al., 2014) but has also been suggested to be poleward-shifting (Yang et al., 2016, 2020; Li et al., 2021, 2022a, b), resulting in a decrease in poleward transport from 28 to 32° S and an increase in eddy activity (and poleward transport) downstream in the EAC southern extension (Li et al., 2021), driving stronger surface warming (Wu et al., 2012; Cetina-Heredia et al., 2014; Malan et al., 2021; Li et al., 2022a, b). These effects, although not completely understood, have been linked to the South Pacific gyre "spinning up" through basin-wide changes in wind stress (Roemmich et al., 2007; Hill et al., 2008; Oliver and Holbrook, 2014; Yang et al., 2020; Li et al., 2022b).

Globally, cross-shore gradients in sea surface temperature trends between the near coast and further offshore ($\sim 150$ km) are common place, including along the eastern coast of Australia (Marin et al., 2021). It is known that continental shelf ocean temperatures in the EAC system are affected by variability in the strength and position of the EAC jet (Archer et al., 2017) and its eddies (Li et al., 2022a), current- and wind-driven upwelling (Roughan and Middleton, 2002, 2004; Schaeffer et al., 2013), vertical and horizontal mixing, and air–sea heat fluxes (Oliver et al., 2021). However, the link between large-scale dynamics and near-coastal temperature is not well understood.

Previous studies have estimated long-term temperature trends on the shelf adjacent to waters affected by the EAC (Thompson et al., 2009; Kelly et al., 2015; Ridgway, 2007; Hill et al., 2008; Holbrook and Bindoff, 1997; Shears and Bowen, 2017). Long-term temperature trends of 0.75 to 1.4 and 1.5 to 2.3 °C per century have been estimated at or close to Port Hacking (near Sydney, 34.1° S) and Maria Island (Tasmania, 42.6° S), respectively, using more than 50 years of (mostly surface) in situ data. More recently, using satellite sea surface temperature data since the 1990s, warming trends of between 1.6 and 4.8 °C per century have been estimated at sites off southeastern Australia between 27 and 42.6° S (Malan et al., 2021). However, to date all temperature trends in the EAC system have been estimated either at or near the surface or using vertically averaged temperatures, and at present little is known of temperature trends below the surface.

It is common to estimate trends in environmental data using linear methods, for example using a least-squares fit (Thompson et al., 2009), a combination of the Mann–Kendall test and Theil–Sen slope estimator (Theil, 1950; Kendall, 1975; Yue et al., 2002), or other statistical methods such as epoch differences (Barnes and Barnes, 2015) or one-way ANOVAs (Kelly et al., 2015). At the two long-term coastal stations influenced by the EAC, Port Hacking and Maria Island, surface temperature change has previously been quantified using linear trends (e.g. Thompson et al., 2009; Shears and Bowen, 2017). Such methods rely on assumptions, for example that the trend is linear or data points are stationary and independent. However, ocean temperature time series are unlikely to have trends that can be approximated well using a straight line over decades (Seidel and Lanzante, 2004; Wu et al., 2007; Cheng et al., 2022) and are often non-stationary (Barbosa, 2011). Recently, Cheng et al. (2022) explored nonlinear methods for quantifying the rate of global ocean heat content change. They found piecewise linear fits and locally weighted scatterplot smoothing worked best when adequate span widths are chosen for estimating multi-decadal trends. Ideally, ocean temperature trends should be estimated without any prior assumptions regarding stationarity and linearity and without using a predetermined functional form.

This study presents estimates of coastal ocean temperature trends at five sites off southeastern Australia spanning a coastline of approximately 2000 km. We use in situ data between the surface and the seafloor over multiple decades. Three sites are situated in the EAC southern extension region, with two of these sites having data extending back more than 7 decades. The remaining two sites are situated upstream of the EAC separation zone inshore of the EAC jet. The impact of the EAC on surface temperature trends varies along the southeastern Australian shelf due to the varying dynamics (Malan et al., 2021), but we know little about temperature trends below the surface, their consistency with surface warming, and how the temperature trends may have varied over time.

We estimate temperature trends using the ensemble empirical mode decomposition (EEMD) method, which is an adaptive and local analysis technique, to derive trends from a time series without the use of predetermined functional forms. In addition, the Theil–Sen slope estimator (TSSE) and Mann–Kendall tests (Theil, 1950; Kendall, 1975; Yue et al., 2002) are used to provide trends for comparison, and the innovative trend analysis (ITA) method (Şen, 2012) is used as a visual tool to explore how temperature distributions have changed over time, highlighting the presence of trends in minimum, middle, and maximum temperatures. We explore temperature trends through the water column, highlighting the complex spatial and vertical structure of ocean warming.

In Sect. 2, we describe the oceanographic sites, their observational data sets and data processing, and briefly the methods of estimating trends. In Sect. 3, we describe the temperature trends in space and time at sites with both short- and long-term records. In Sect. 4, our results are discussed in the context of the local and broad-scale dynamics, the pros and cons of the methodologies are explored, and a comparison is made between our results and previous studies. We conclude our study in Sect. 5.

## 2 Data and methods

### 2.1 Oceanographic sites and their data sets

We use temperatures at two long-term oceanographic sites (Fig. 1) starting in the 1940–1950s and continuing to the present. One site is located just south of Sydney at Port Hacking ($\sim 34.1°$ S) in 110 m of water, downstream of the typical EAC separation point (30 to 32.5° S). The other site is off eastern Tasmania at Maria Island ($\sim 42.6°$ S) in 90 m of water at the southern end of the EAC extension region. In 2009, these sites were incorporated into the Integrated Marine Observing System (IMOS) National Reference Station (NRS) network (Lynch et al., 2014) and have been referred to as NRSPHB and NRSMAI, respectively. The records contain data collected weekly to monthly via in situ boat-based sampling and 5 min to hourly electronic sensor data. At Port Hacking, mooring measurements from a nearby site, PH100, are used. Sampling is at multiple depths between the near surface (0–

1 m) and the near bottom (100 m). Here we use nearly all available temperature data at both sites from the surface to the bottom since the records commenced as shown in Fig. 2. The long-term temperature data from these sites have been packaged into validated and tested NRS data products as described by Roughan et al. (2022a), which we use here updated to the end of 2022.

The Port Hacking and Maria Island sites were chosen for two reasons. (1) The sites are long-term, containing over 70 years of temperature data and enabling long-term trend detection. (2) One site is close to the EAC separation region, while the other is further downstream in the EAC southern extension region. Hence, we can compare trends at locations with contrasting oceanographic conditions. At Port Hacking there are two long-term sites (50 and 100 m), but only the 100 m site has moored measurements; hence we use data from the 100 m site.

In addition, we use temperature records from the more recently occupied sites, including the NRS North Stradbroke Island 63 m depth mooring ($\sim 27.3°$ S), the Coffs Harbour 100 m depth mooring ($\sim 30.25°$ S), and the Batemans Marine Park 120 m depth mooring ($\sim 36.2°$ S) (Fig. 1). Each site has approximately a decade of temperature data (Fig. 2), with the longest record available at Coffs Harbour (late 2009 to present) (Roughan et al., 2013). The temporal sampling of the moored sensors ranges from 5 min to hourly, with sensors located at multiple depths between the shallowest depth, typically at 8 to 20 m, and the bottom.

At these newer short-term sites we use the mooring aggregated long time series products developed by the Australian National Mooring Network (ANMN) and the Australian Ocean Data Network (AODN) (IMOS, 2021a, b, c), which combine multiple-deployment temperature files into one aggregated file per site. Additionally, as Roughan et al. (2022a) determined that satellite data can be used to augment the existing mooring data after 2012, we combine surface satellite data with the subsurface long time series products at these three sites, as well as at the two long-term sites Port Hacking and Maria Island, similarly to the method described by Roughan et al. (2022a).

All temperature data used in this study have been quality-controlled. The historical bottle data were initially quality-controlled by the Commonwealth Scientific and Industrial Research Organisation (CSIRO) with evolving practice over time, while IMOS CTD (conductivity–temperature–depth) and mooring data collected since 2008–2009 were quality-controlled using standardised IMOS procedures (Ingleton et al., 2014; Lara-Lopez et al., 2017; AODN, 2023). For satellite data, quality level flags $>= 4$ were used to select temperature data. Additional quality control checks were performed for all data sets, as described by Hemming et al. (2020), and further information on data quality control is described by Roughan et al. (2022a).

Before 2009, the long-term data sets include temperature measured with reversing thermometers. These temperatures have an estimated accuracy of better than $\pm 0.02$ °C. The long-term data sets also include electronic CTD profiles since 1997 and 2009 at Port Hacking and Maria Island, respectively. Typically Sea-Bird Electronics sensors (SBE 25, SBE 17plus, SBE 19plus) have been used for CTD profiles. The SBE 19plus sensors have an initial accuracy of better than $\pm 0.005$ °C.

Mooring data used for both the long-term (Port Hacking and Maria Island) and short-term (North Stradbroke Island, Coffs Harbour, and Batemans Marine Park) data sets consist of temperatures measured by various electronic sensors (e.g. Aquatec AQUAlogger 520T/520TP, WET Labs CE2 water quality meters, SBE 37). The initial accuracy for most moored sensors is $\pm 0.002$ °C, but for Aquatec loggers it is $\pm 0.05$ °C. The temperature sensors used for electronic CTD profiles and for the moorings have been calibrated annually at a CSIRO calibration facility in Hobart since 2009. Where possible, temperatures observed using the various data platforms close in time and space to one another were matched and compared, and our analysis did not indicate any major systematic biases. CE3

## 2.2 Gap filling and averaging

It is important to consider the data gaps in the temperature time series prior to estimating trends. For our analysis, monthly binned temperatures were used at all sites (Fig. 2) that were calculated from the data sets described above. For investigating trends using the TSSE and ITA methods (see the following section), the monthly binned temperatures were deseasonalised by subtracting the monthly temperature climatology. An example of what the deseasonalised temperature data look like alongside estimated trends is provided in Fig. A1 in Appendix A.

The time series used here had some gaps of days to years, as identified by Roughan et al. (2022a) (see their Fig. 2 for Port Hacking and Maria Island), depending on site location, depth, and retrieval method. To limit the effect of data gaps on the trend estimates, gaps were filled prior to using trend methodologies. We used a synthetic temperature time series created from a combination of the mean climatology, a long-term signal based on deseasonalised temperature, and simulated red noise. More information relating to the gap-filling method can be found in Appendix A, and examples of gap-filled monthly data are shown in Fig. 2.

## 2.3 Detecting trends

For an ocean temperature time series, the underlying variability and trend are likely to be non-linear and non-stationary (Barbosa, 2011). For that reason, we use the ensemble empirical mode decomposition (EEMD) method to determine trends without relying on prior assumptions (Wu and Huang, 2009; Huang et al., 1998). The EEMD method has been used in numerous environmental studies (e.g. Wu et al., 2007;

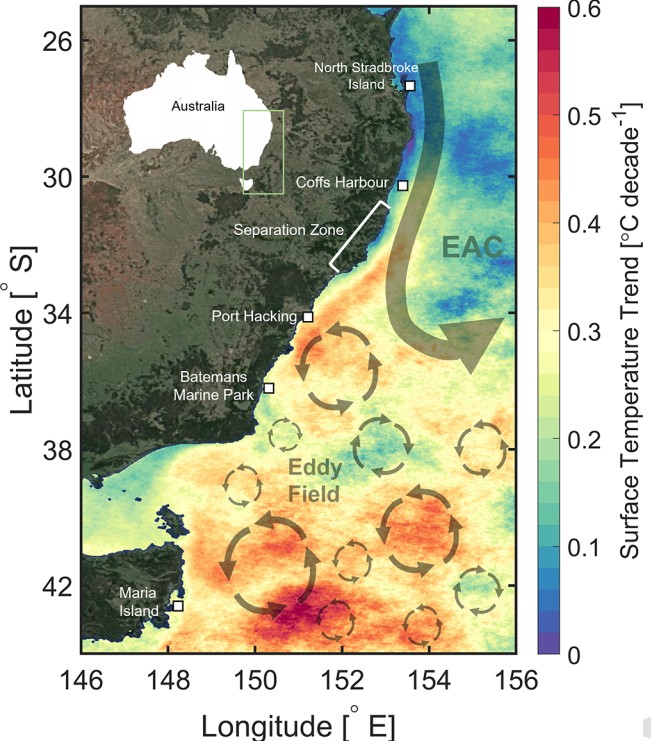

**Figure 1.** The locations of the five oceanographic sites off south-eastern Australia, from north to south: North Stradbroke Island, Coffs Harbour, Port Hacking, Batemans Marine Park, and Maria Island. The decadal surface temperature trends from the SST (sea surface temperature) Atlas of Australian Regional Seas (SSTAARS) using data between 1992 and 2016 (Wijffels et al., 2018) are plotted, with broad-scale circulation patterns including the East Australia Current (EAC) and its associated downstream eddy field superimposed on top. Satellite and map information sourced from ©Google Maps 2022.

Chen et al., 2017; Ji et al., 2014; Molla et al., 2006) and is described in detail in Appendix B.

The Mann–Kendall trend test was used alongside the TSSE method to estimate linear trends for comparison with the non-linear EEMD trends. The Mann–Kendall trend test detects the presence of a significant trend in a time series using rank (Mann, 1945; Kendall, 1975) and has been used in numerous environmental studies (Atta-ur-Rahman and Dawood, 2017; Praveen et al., 2020; Douglas et al., 2000). The Mann–Kendall test requires independent data, although in reality most time series are autocorrelated (Hamed and Rao, 1998). As described by Von Storch and Navarra (2013), the presence of positive serial correlation in a stochastic time series can increase the probability of detecting a false-positive trend. To account for serial correlation, we used the trend-free pre-whitened version of the Mann–Kendall trend test (Yue and Wang, 2002; Yue et al., 2002). From here on we will refer to the combined Mann–Kendall TSSE trend method as TSSE.

The innovative trend analysis (ITA) method (Şen, 2012) is useful for highlighting changes over time in minimum, middle, and maximum temperatures between two distributions and has been used in environmental science (e.g. Sanikhani et al., 2018; Mohorji et al., 2017). A time series $x(t)$, which in our case is a monthly gap-filled temperature (deseasonalised, Fig. A1) anomaly time series, is first split into two equal segments representing the same time period length, and the first ($x_i$) and last ($y_i$) segments are sorted into ascending order. Segments $x_i$ and $y_i$ are then plotted against each other alongside a $1:1$ line, with $x_i$ typically on the $x$ axis. If there is no trend, the data points will appear close to the $1:1$ line, whereas if there is a positive or negative trend, the data points will appear above or below the $1:1$ line, respectively. A constant trend across the temperature distribution will appear parallel to the $1:1$ line, whereas varying trends will not.

We compare the non-linear temperature trends estimated using the EEMD method with those estimated using the TSSE method. We make this comparison for the following three reasons. (1) The TSSE method is a linear method which is more commonly used than the non-linear EEMD method. (2) We can therefore easily compare our TSSE trends with linear trends estimated in previous studies. (3) When considering most of the sites, these are the first trend estimates below the surface. Therefore, we can provide estimates at the sites using the different methods for easier comparison in the future and can also highlight the effect of methodology choice and their assumptions in estimating trends.

Additionally, to highlight how the temperature trends have evolved over time at the long-term sites and to allow for temporal contextualisation for other shorter studies, we show the EEMD trends for each decade on record. We take the mean of the first-order temporal monthly derivative of the EEMD temperature trend ($R(t)$) for each decade multiplied by 120 to reveal the mean decadal trends.

Although data are available since 1944 at Maria Island, we estimate long-term temperature trends at this site between 1953 and 2022 for consistency with Port Hacking. Further, we use temperature data from 2012 onward at sites with short-term temperature records for consistency over depth as the satellite surface data that we use start in 2012.

## 3 Results

### 3.1 Multi-decadal trends

The temperature trends between 1953 and 2022 are estimated at Port Hacking and Maria Island revealing considerable differences between the two sites (Figs. 3, 4). Overall, warming at Port Hacking is surface- and bottom-intensified, while further south at Maria Island, warming is more consistent over depth. Trends are accelerating over decades at both Port Hacking and Maria Island, particularly at the surface, al-

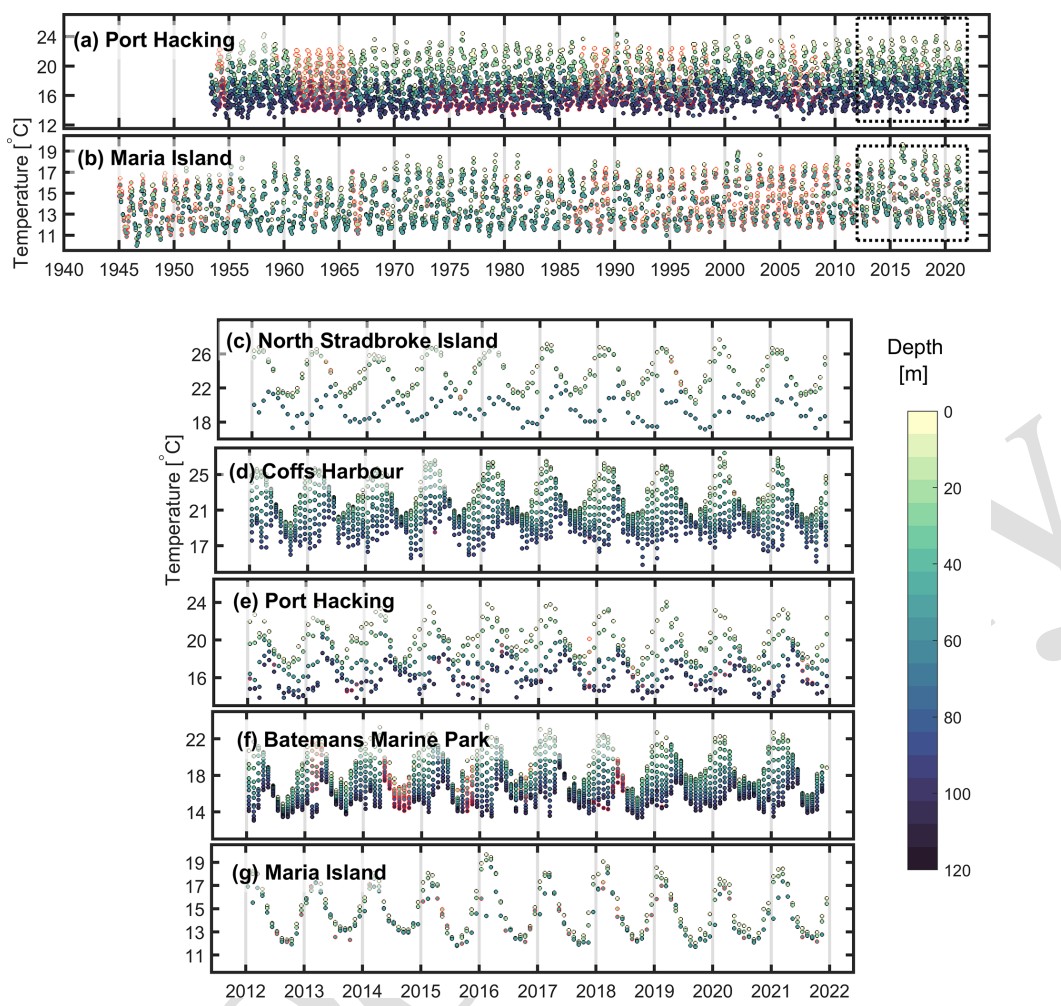

**Figure 2.** Multi-decadal gap-filled temperature time series at multiple depths at **(a)** Port Hacking incorporating PH100 mooring data and **(b)** Maria Island. The same data at **(e)** Port Hacking and **(g)** Maria Island between 2012 and 2022 TS1 and the shorter gap-filled time series at **(c)** North Stradbroke Island, **(d)** Coffs Harbour, and **(f)** Batemans Marine Park between 2012 and 2022 TS2. Data points are coloured by depth and surrounded by a red edge if gap-filled. Also note the different *y*-axis limits varying by site. The dashed boxes in **(a)** and **(b)** indicate the data shown in **(c)** and **(d)**.

though the uncertainty (Figs. 3, 4) must also be considered when evaluating these EEMD trends.

The EEMD temperature trends at Port Hacking are estimated at depths of 2, 22, 50, 77, and 99 m. Results show that at the surface and bottom the EEMD trends are statistically significant and have been accelerating from the late 1990s (Figs. 3a, 4a), relative to earlier decades. At most depths EEMD trend acceleration is detected from the 1970s, although it is not statistically significant until the 1990s as determined by the methodology described in Appendix B. Warming rates are highest at the surface off Port Hacking with rates $\geq 0.2\,°C$ per decade over the last 3 decades, while the EEMD trends at mid-depths are lower and are not statistically significant. Surprisingly over the last 2 decades at 99 m depth, waters have warmed $\geq 0.12\,°C$ per decade, and during the 2010s at a depth of 77 m, waters significantly warmed $0.18\,°C$ per decade.

The Maria Island EEMD trends at depths of 2 and 20 m have been statistically significant since the mid-2000s (Figs. 3b, 4b). The results show that the Maria Island coastal waters have warmed consistently since the 1950s in the top 20 m of the water column relative to Port Hacking, with similar time period average EEMD trends (0.16–0.19 °C per decade, Fig. 4b) estimated at the site at 2 and 20 m depth. The Maria Island 2 and 20 m depth trends have accelerated, similarly to surface and bottom trends at Port Hacking. In contrast, the 50 m trend accelerated between the 1950s and 1990s but decelerated between the 1990s and 2010s.

The uncertainty for EEMD trends at both Port Hacking and Maria Island at any given time is shown in Fig. 3. The

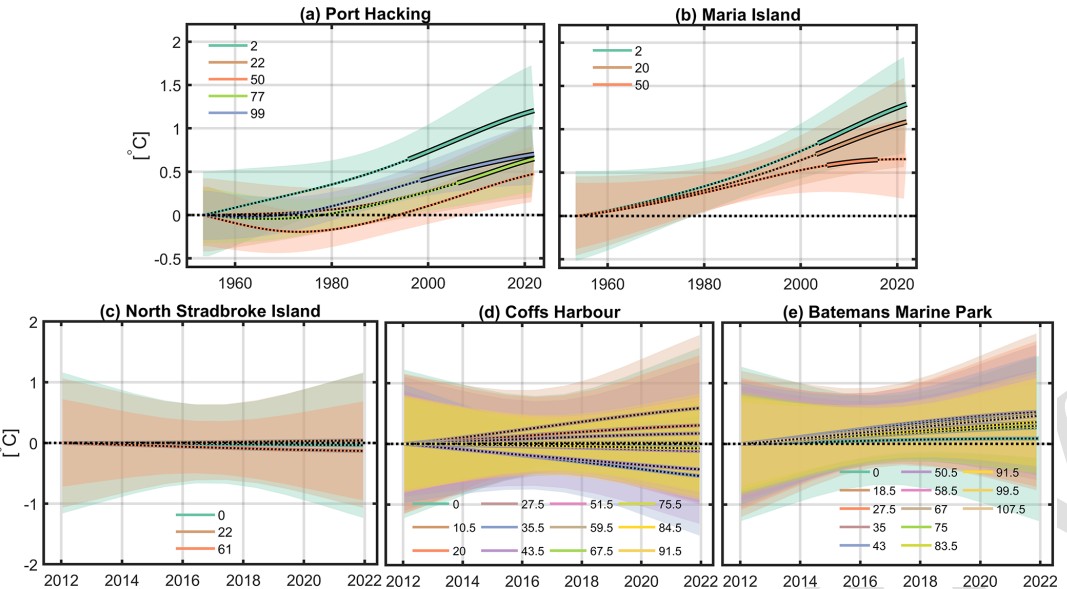

**Figure 3.** The temperature ensemble empirical mode decomposition (EEMD) trends at each of the five sites: **(a)** Port Hacking, **(b)** Maria Island, **(c)** North Stradbroke Island, **(d)** Coffs Harbour, and **(e)** Batemans Marine Park. Each coloured line represents a depth level (metres) at each of the sites, as indicated in the corresponding legends. The uncertainty for each depth level estimated using the downsampling method is represented by the shaded area with the colour corresponding to the lines. Significant trend periods are represented by a filled line with a black outline. Insignificant trend periods are indicated by dashed lines. Note the difference in y- and x-axis limits between **(a)**–**(b)** and **(c)**–**(e)**.

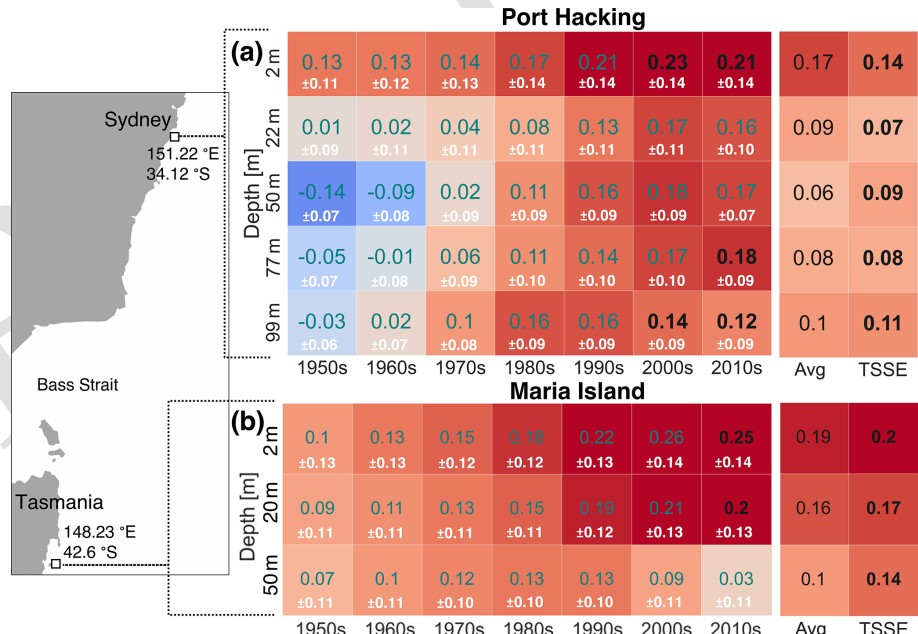

**Figure 4.** Temperature trends for multiple depths at **(a)** Port Hacking and **(b)** Maria Island. Statistically significant (black bold text) and statistically insignificant (grey text) average EEMD trend rates per decade (°C per decade) and the total time average over all decades ("Avg", black text) are shown. The uncertainty for each decade is listed beneath the average EEMD trend rates as white text. The statistically significant Theil–Sen slope estimator (TSSE) trend estimates (°C per decade) using data over the whole time period are also shown for each site. The locations of the sites are shown in the left panel. The total time-averaged EEMD estimates use both statistically significant and insignificant trend rates over the whole time period and thus are taken as insignificant estimates. Decade trends are considered significant if 75 % or more of the trend during the selected decade are outside the 95 % confidence bounds.

uncertainty at the surface is approximately 0.5 °C close to the time period edges (1953 and 2022) and approximately 0.3 °C between the 1980s and 2000s. When considering the whole time series, the uncertainty for the decadal trends is shown in Fig. 4 and is approximately ±0.07 to 0.14 °C per decade. In general, uncertainty below the surface is lower than at the surface, and uncertainty decreases with depth, with the most robust results at the bottom. We tested the uncertainty estimates (as discussed and shown in Appendix C and Fig. A3, respectively), and we are confident that the accelerating rates of warming presented are robust.

Trends estimated using both the EEMD and TSSE (combined Mann–Kendall and Theil–Sen slope estimator) methods are compared at Port Hacking and Maria Island (columns labelled "Avg" and "TSSE" for the EEMD and TSSE methods, respectively, in Fig. 4). The high Port Hacking trends at the top and at the bottom of the water column and the depth-consistent warming at Maria Island relative to Port Hacking are generally reflected in both the EEMD and TSSE trends, and the TSSE trends are statistically significant at all depths.

The ITA analysis (Fig. 5) confirms the EEMD and TSSE trend results that temperatures are generally increasing at both sites. However, the two long-term sites also show some differences. The trends vary for minimum, middle, and maximum temperature anomalies. At Port Hacking, the warmest temperature anomalies have increased more over time than the lowest temperature anomalies, clearest at the surface and at the bottom. Trends also vary over depth at this site, with a decreasing trend observed for minimum temperatures at 22 m depth. At Maria Island, there is consistent warming for all temperature anomalies at all three depths relative to Port Hacking. Some maximum and lower middle temperature anomalies have warmed more than other temperature anomalies across the distribution.

## 3.2 Short-term trends

To provide spatial context to the long-term trends at Port Hacking and Maria Island, we estimate temperature trends at three sites between 2012 and 2022 (North Stradbroke Island, Coffs Harbour, and Batemans Marine Park) positioned along the coastline adjacent to the EAC (Fig. 1). We do not consider these trends representative of longer periods (e.g. 30 years or more), as inter-decadal variability will likely play a role. Rather we include these trends as preliminary summaries of temperature change over the period in which we have data, and we expect that these trend estimates will strengthen over time and become statistically significant as we collect more data at these sites. Not surprisingly, we find that the majority of the trends in the shorter time series are not statistically significant and have higher uncertainty than the long-term trends (Figs. 3c–e, A2). In the northern EAC jet region at the sites North Stradbroke Island and Coffs Harbour (Fig. 3c, d), there is a mixture of depth-dependent warming or cooling trends. North Stradbroke Island shows slight warming

at 22 m and cooling elsewhere, whilst Coffs Harbour shows low rates of warming closer to the surface, alongside cooling subsurface waters. Further south downstream of Port Hacking at the Batemans Marine Park site CE4 (Fig. 3e) in the EAC extension region we see insignificant EEMD warming trends but some significant TSSE warming trends (Fig. A2) that vary in intensity between the surface and the bottom. However, the trends at Batemans Marine Park can be considered relatively consistent over depth when compared with the other short-term sites.

## 4 Discussion

### 4.1 Contextualising the observed trends

#### 4.1.1 Surface warming

Here we discuss the long-term trends observed at Port Hacking (34.1° S) and Maria Island (42.6° S) and place them in the global and regional context. It is known that the East Australian Current is warming rapidly (Wu et al., 2012), with the fastest warming observed between Port Hacking and Maria Island (Malan et al., 2021) driven by changes in wind forcing and eddy generation (Li et al., 2022b).

Near-surface ocean temperatures at Port Hacking and Maria Island have warmed at a faster rate (0.14 to 0.2 °C per decade) than the global surface (land and ocean) average (approximately 0.14 °C per decade; Rohde and Hausfather, 2020). Mean near-surface temperature estimated using the EEMD method is now approximately 1.2 °C warmer at Port Hacking and Maria Island than it was in 1953, approximately 0.2 °C more than the total global surface (land and ocean) average temperature change since 1953 (Rohde and Hausfather, 2020).

Malan et al. (2021) showed that the shelf warming in the EAC southern extension is almost solely advection-driven; hence we suspect that near-surface temperatures at Port Hacking are predominantly driven by the increased poleward penetration of the EAC (and its eddies) (Li et al., 2022b), as well as atmospheric changes. Increased poleward-penetration of western boundary currents, such as the EAC, is driving a redistribution of heat, bringing more warm water to southern latitudes (Hu et al., 2015; Li et al., 2022b). The warming at Maria Island, particularly since the 1980s, is also consistent with Kelly et al. (2015), who showed that the amount of EAC extension water at this site rapidly increased over the same time period.

Our results show evidence of accelerating warming trends (Figs. 3 and 4) and that waters have warmed the most near the surface at both Port Hacking and Maria Island. The Port Hacking and Maria Island EEMD warming rates at 2 m depth have accelerated over time (0.2 to 0.25 °C per decade during the 2010s), with uncertainty of ±0.11 to ±0.14 °C per decade (Fig. 4).

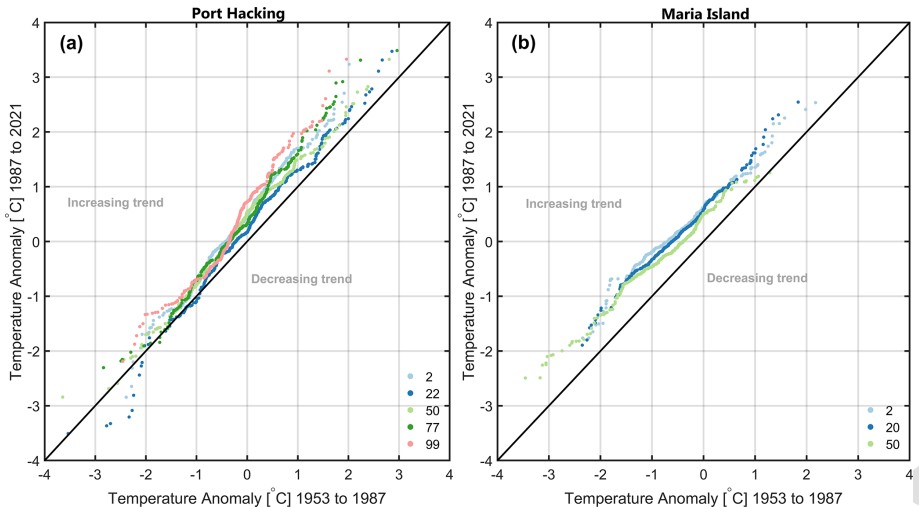

**Figure 5.** Innovative trend analysis (ITA) plots for temperatures with the seasonal cycle removed from top to bottom (depth in metres, coloured scatter points) at **(a)** Port Hacking and **(b)** Maria Island. If scattered temperature anomalies are in the top or bottom triangle, there is either an increasing or a decreasing trend, respectively, as labelled.

The accelerating temperature trend near the surface at Port Hacking and Maria Island is consistent with previously reported surface trends. For example, Thompson et al. (2009) estimate a trend of 0.07 and 0.20 °C per decade at Port Hacking and Maria Island, respectively, using data between 1953 and 2005. While Kelly et al. (2015) estimate higher trends of 0.14 and 0.21 °C per decade when using similar data sets extended from 1953 to 2012 and 1950 to 2012 at Port Hacking and Maria Island, respectively. Shears and Bowen (2017) also present possible acceleration at Maria Island providing temperature trends of 0.2 °C per decade from 1946 to 2016 and 0.32 °C per decade from 1982 to 2016, respectively, keeping in mind that these trends likely have considerable uncertainty and overlap with one another. This local trend acceleration is consistent with a global acceleration in ocean heat content since the 1980s (Cheng et al., 2022).

### 4.1.2  Subsurface warming

The warming at depth at Port Hacking is also noteworthy, commencing in the 1970s. The bottom warming is unlikely to be the result of increased wind-driven vertical mixing as the average mixed-layer depth between 2006 and 2017 at Port Hacking was approximately 22 m (Van-Ruth et al., 2020) and mid-waters have warmed at slower rates than the surface and bottom on average over the entire time period. Instead, we suspect that bottom waters may have warmed through modifications to upwelling (which typically drives the coldest bottom temperatures during the summer season; Wood et al., 2013; Roughan et al., 2022a), with additional circulation influences as the EAC warms and becomes more eddying offshore (Malan et al., 2021; Li et al., 2022a).

Li et al. (2022b) show (in their Fig. 5b) a poleward shift in the easterly (westward) winds in Southern Hemisphere 10 m zonal mean ocean surface winds at 34° S between 1993 and 2020. They also show that zonal mean winds at this latitude are westerly (eastward). These changes in zonal winds could potentially point towards a suppression of upwelling-favourable winds, keeping in mind that an analysis of meridional winds is also required. If there was a decrease in upwelling, it would mean lower nutrient concentrations at the bottom (Roughan and Middleton, 2002). However Thompson et al. (2009) showed an increasing surface nitrate trend at Port Hacking between 1953 and 2005 which might instead suggest an increase in upwelling (noting their study period ended in 2005). Alternatively, the increased bottom temperatures could be a consequence of offshore warming and mixed-layer deepening, where the source of the upwelled water has warmed. At Maria Island, waters have warmed consistently over the upper 20 m of the water column, relative to Port Hacking. This is likely because the site itself is far less stratified than Port Hacking with lower seasonal temperature variability (Thompson et al., 2009; Roughan et al., 2022a).

### 4.2  Pros and cons of EEMD and TSSE methodologies

The EEMD method is useful as it shows rates of warming over time, but the uncertainty associated with the EEMD trends has to be considered. We suspect that a large portion of the higher uncertainty that we estimate close to the time series start and end points (Fig. 3) is due to the methodology, rather than due to the temperature time series. This is because the EEMD method suffers from edge effects (Stallone et al., 2020). We explored the extension algorithm provided by Stallone et al. (2020) for reducing edge effects, but sensitivity tests indicated that in our case it was better to use the original non-extended time series.

**Table 1.** A comparison of trends estimated in this study and in other studies using observations at Port Hacking and Maria Island. We compare trends at and below the surface for different time periods. The trend time period, method, and data platforms used are also shown. The trends estimated by Wijffels et al. (2018) correspond to those shown in Fig. 1, and trend estimates taken from studies denoted with "a" are not from the exact location of the sites but instead from the approximate area. The trend estimated by Holbrook and Bindoff (1997), denoted with "b", uses vertically averaged temperature changes for the upper 100 m of the water column at 43° S, 149° E. MBT: mechanical bathythermograph, XBT: expandable bathythermograph.

| Site, bottom depth | Depth (m) | Study | Time period | Trend (°C per decade) | Trend method | Data platform |
|---|---|---|---|---|---|---|
| Port Hacking 110 m | Surface | Kelly et al. (2015) | 1953–2012 | 0.14 | Linear | Bottle, CTD |
| | Surface | This study | 1953–2022 | 0.14 | Linear (TSSE) | Bottle, CTD, mooring, satellite |
| | Surface | This study | 1953–2022 | 0.18 | EEMD | Bottle, CTD, mooring, satellite |
| | Surface | Thompson et al. (2009) | 1953–2005 | 0.07 | Linear | Bottle |
| | Surface | Wijffels et al. (2018) [a] | 1992–2016 | 0.33 | Linear | Satellite |
| | Surface | Foster et al. (2014) [a] | 1993–2013 | 0.16 | Linear | Satellite |
| | Surface | Malan et al. (2021) [a] | 1993–2017 | 0.48 | Linear | Satellite |
| | Surface | This study | 2008–2019 | 0.8 | Linear (TSSE) | Satellite, mooring, CTD |
| | Surface | This study | 2008–2019 | 0.86 | EEMD | Satellite, mooring, CTD |
| | 22 | This study | 1953–2022 | 0.07 | Linear (TSSE) | Bottle, CTD, mooring, satellite |
| | 22 | This study | 1953–2022 | 0.09 | EEMD | Bottle, CTD, mooring, satellite |
| | 22 | This study | 2008–2019 | 0.77 | Linear (TSSE) | Mooring, CTD |
| | 22 | This study | 2008–2019 | 0.99 | EEMD | Mooring, CTD |
| | 23 | Malan et al. (2021) [a] | 2008–2019 | 0.9 | Linear | Mooring |
| | 99 | This study | 1953–2022 | 0.11 | Linear (TSSE) | Bottle, CTD, mooring, satellite |
| | 99 | This study | 1953–2022 | 0.1 | EEMD | Bottle, CTD, mooring, satellite |
| Maria Island 90 m | Surface | Ridgway (2007) | 1944–2002 | 0.23 | Linear | Bottle |
| | Surface | Hill et al. (2008) | 1944–2004 | 0.22 | Linear | Bottle |
| | Surface | Thompson et al. (2009) | 1944–2005 | 0.2 | Linear | Bottle |
| | Surface | Shears and Bowen (2017) | 1946–2016 | 0.2 | Linear | Bottle, CTD |
| | Surface | Kelly et al. (2015) | 1950–2012 | 0.21 | Linear | Bottle, CTD |
| | Surface | This study | 1953–2022 | 0.2 | Linear (TSSE) | Bottle, CTD, mooring, satellite |
| | Surface | This study | 1953–2022 | 0.19 | EEMD | Bottle, CTD, mooring, satellite |
| | Surface | Shears and Bowen (2017) | 1967–2016 | 0.16 | Linear | Bottle, CTD |
| | Surface | Shears and Bowen (2017) | 1982–2016 | 0.32 | Linear | Bottle, CTD |
| | Surface | Wijffels et al. (2018) [a] | 1992–2016 | 0.26 | Linear | Satellite |
| | Surface | Foster et al. (2014) [a] | 1993–2013 | 0.38 | Linear | Satellite |
| | Surface | Malan et al. (2021) [a] | 1993–2017 | 0.41 | Linear | Satellite |
| | Surface | This study | 2008–2019 | 0.5 | Linear (TSSE) | Satellite, mooring, CTD |
| | Surface | This study | 2008–2019 | 0.58 | EEMD | Satellite, mooring, CTD |
| | 20 | This study | 1953–2022 | 0.18 | Linear (TSSE) | Bottle, CTD, mooring, satellite |
| | 20 | This study | 1953–2022 | 0.16 | EEMD | Bottle, CTD, mooring, satellite |
| | 20 | This study | 2008–2019 | 0.55 | Linear (TSSE) | Mooring, CTD |
| | 20 | This study | 2008–2019 | 0.76 | EEMD | Mooring, CTD |
| | 20 | Malan et al. (2021) | 2008–2019 | 1.03 | Linear | Mooring |
| | 0–100 | Holbrook and Bindoff (1997) [a,b] | 1955–1988 | 0.15 | Linear | MBT and XBT casts |
| | 50 | This study | 1953–2022 | 0.14 | Linear (TSSE) | Bottle, CTD, mooring, satellite |
| | 50 | This study | 1953–2022 | 0.1 | EEMD | Bottle, CTD, mooring, satellite |

To test whether the estimated EEMD uncertainty was predominantly a result of the methodology, we compared the EEMD trend and its uncertainty with a piecewise linear fit trend and its uncertainty (Fig. A3), following the methodology described in Appendix C. The piecewise linear fit results also confirm that the warming trend is accelerating over time and has lower uncertainty than the EEMD trend uncertainty. Further, the EEMD trend is within the uncertainty range of the piecewise linear fit trend. This comparison suggests that, despite the higher uncertainty, the EEMD method is useful for identifying a meaningful accelerating trend.

The selected time period also has a considerable effect on temperature trends. It is known that linear trends are sensitive to the time period choice, as we demonstrate in Sect. 4.3 and for the piecewise linear fit in Fig. A3. However, we find EEMD trends are also sensitive to the time period choice. For example, if we estimate 2 m depth EEMD trends at Port Hacking using temperatures between 1953 and 2019 instead

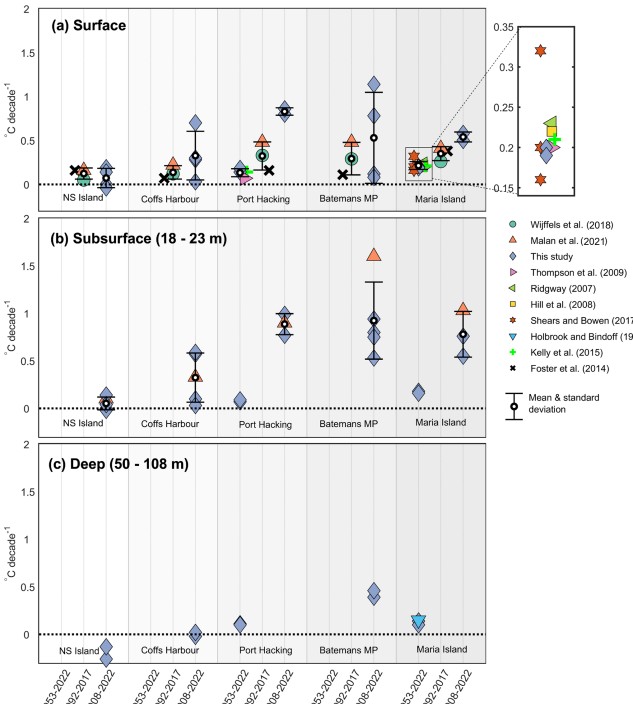

**Figure 6.** A comparison of temperature trends estimated in this study with trends estimated in other studies, **(a)** at the surface; **(b)** at the subsurface; and **(c)** deep for the sites North Stradbroke (NS) Island, Coffs Harbour, Port Hacking, Batemans Marine Park (MP), and Maria Island. Trends are organised as the general time periods of 1944–2020, 1992–2017, and 2010–2020, but their exact lengths vary and are listed in Table 1 and in Table S1 in the Supplement TS3. The means and standard deviations for each time period and depth are overlain on top of individual trend estimates, and because of overlap, there is a close-up of the trends shown in **(a)** between 1944 and 2020 at Maria Island. Note that all of the EEMD trends shown between 2008 and 2022 in this study are not significant and have estimated uncertainty of between 0.07 and 2.3 °C per decade dependent on site location and depth. Further, different time series lengths were used for the short-term trends as follows: North Stradbroke Island, Coffs Harbour (2010–2019, 2012–2022), and Batemans Marine Park (2011–2019, 2012–2022).

of between 1953 and 2022, we derive a total trend change of 1.5 °C instead of the 1.2 °C shown in Fig. 3, keeping in mind that this difference of 0.3 °C is within the uncertainty in the EEMD trend estimate (approximately 0.5 °C).

For our data sets, we find that the TSSE method is suitable for approximating the overall trends as they compare well with the time-averaged EEMD trends (assuming that these are closest to reality). The TSSE method is also simpler; hence this method will be faster and less resource-intensive relative to the EEMD method. However, the TSSE trends are not useful for deriving varying warming rates over long periods, and hence if this is an objective, the EEMD method should instead be considered.

## 4.3 Comparison with trends from other studies

In order to provide a comprehensive record of temperature trends in our region, we compare our results with previous studies that have investigated some aspect of temperature trends at or close to the sites used in this study (Wijffels et al., 2018; Malan et al., 2021; Thompson et al., 2009; Ridgway, 2007; Hill et al., 2008; Shears and Bowen, 2017; Holbrook and Bindoff, 1997; Kelly et al., 2015; Foster et al., 2014). Each of these studies have used linear methods to estimate the trends, and most studies have used surface data only. While our study has explored non-linear trends between the surface and the bottom.

A comparison with previously published temperature trends (Fig. 6, Table 1) supports our findings that trend magnitudes are depth-dependent and vary across latitude, further highlighting that trends are sensitive to the time period chosen. The temperature trend rate is often higher and with higher uncertainty when the record is shorter and includes more recent data, relative to those estimated using longer time periods. This further confirms the accelerating warming that we observe over time.

From these studies, three looked at temperature trends below the surface at or close to the long-term sites: Malan et al. (2021), Holbrook and Bindoff (1997), and Thompson et al. (2009). While Malan et al. (2021) estimated subsurface trends at depths of approximately 20 m (their Table S2 TS4), Holbrook and Bindoff (1997) used depth-averaged temperature changes for the upper 100 m of the water column some distance away from Maria Island. Thompson et al. (2009) used depth-averaged temperatures to estimate the seasonal trends (a trend for each month of the year) and hence cannot directly be compared with our annual trends.

When using subsurface temperature data between 2010 and 2019 at North Stradbroke Island and Coffs Harbour and between 2008 and 2019 at Port Hacking, our trends are similar to those presented by Malan et al. (2021), keeping in mind that at Port Hacking their trends were estimated at another site approximately 25 km to the northeast. We find that our 50 m Maria Island trends (Fig. 6) agree relatively well with those estimated by Holbrook and Bindoff (1997) between 1955 and 1988 at a rate of approximately 0.15 °C per decade. This similarity exists even though we use different time periods (e.g. 1953 to 2022), data platforms, and depth ranges, suggesting that the rate of change has been relatively constant during this time. This implies that the fast-changing regional dynamics at the site play less of a role in long-term temperature change and points to large-scale drivers on longer timescales. Despite the low number of studies that use subsurface temperature data, these comparisons further highlight the depth dependency of trends, suggesting that we need to consider the full water column and local dynamics when characterising regional environmental change.

## 5   Conclusions and outlook

We have characterised coastal ocean temperature trends at five shelf locations spanning approximately 2000 km of the southeastern Australian coastline adjacent to a major western boundary current. We use the EEMD method to estimate non-linear trends that provide the time-varying rates of change, keeping in mind the estimated uncertainty. Using this method, we estimate an acceleration in the near-surface trends at Port Hacking and Maria Island consistent with trends seen globally. This acceleration is related to modifications of the EAC system and the atmosphere under anthropogenic warming.

Our results off Port Hacking show that temperature trends are highest at the surface and at the bottom, with temperature trends here varying over time at different rates to mid-waters. Temperature trends at Port Hacking vary more over depth than trends at Maria Island, and rates at both sites vary over time. We discuss the importance of regional dynamics in driving these temperature trends.

Marine species that inhabit coastal waters are expected to change or adapt as a response to rising temperature and extremes (Vergés et al., 2014; Niella et al., 2020; Smith et al., 2022). As marine species are often not confined to the surface waters, it is therefore important to understand how temperature will change over time throughout the water column. For example, coastal regions will likely undergo tropicalisation of their ecosystems (Vergés et al., 2014); hence understanding temperature change at depths where species live will be vital for understanding ecosystem response to warming. Our study is the first to explore temperature change beneath the surface at multiple depth levels at these sites and will aid future studies on the potential impacts of temperature change on subsurface marine species. Additionally, understanding trend velocity may provide context for environmental tipping points where marine species are impacted beyond their rates of recovery.

We compare our non-linear EEMD trends with linear trends estimated using the TSSE method. When considering the long periods, we find that linear trends approximate the temperature trends well over the entire time period but that they are prone to under- or overestimate the trend during selected shorter time periods.

Future studies may consider using the EEMD (or similar) method to estimate temporal variability in warming trends over the larger Tasman Sea region using satellite sea surface temperature measurements to complement the work done by Malan et al. (2021), Wijffels et al. (2018), and others using linear trends. Estimating the time-varying rates of change using data over the satellite record will be useful in determining how and where warming (and potentially cooling) has accelerated or plateaued over time. Keeping in mind that insights gained from doing this will be limited to the surface, we show that the subsurface trends and therefore overall shelf heat content can vary from those at the surface. We have not investigated whether the trends are homogeneous throughout the year, although there is evidence to suggest that trends may vary between seasons (Thompson et al., 2009; Shears and Bowen, 2017), and this will be studied in future work.

Our results show that subsurface information is important for understanding the full extent of environmental change through the water column. We also show that considering a range of site locations is also important, as warming rates are complex and heterogeneous along the length of a coastline influenced by a western boundary current.

## Appendix A:  Gap filling

The temperature time series used here had gaps of some days to years, as identified by Roughan et al. (2022a), depending on site location, depth, and retrieval method. For example, the largest gap is a full-depth data gap of approximately 6 years (1960 to 1966) at Port Hacking out of the approximately 69 years of data, and there were many smaller gaps ranging from a few days to a few months. The presence of these gaps at certain times of the year or during certain years/decades would likely lead to a biased trend estimate. For example, after accounting for seasonality, temperature variability in summer is expected to be quite different to that in winter. Further, seasonally corrected temperatures are expected to vary inter-annually. Therefore, data gaps that dominate a particular season or an extended period of time are expected to have an effect on the trend, and gaps were filled to limit this potential effect.

Synthetic data with the same temporal resolution as the binned real data were created over the same time period as the original time series using real data characteristics. These synthetic data were created using a combination of the mean climatology, an inter-annual or inter-decadal signal (depending on the length of the data set) based on real deseasonalised temperatures, and simulated red noise (integration of white Gaussian noise). This red noise had a serial correlation and standard deviation similar to the original time series. These synthetic data were then used to fill gaps in the time series.

To test the effectiveness of this methodology, we simulated gaps of between 10 % and 50 % of real data points missing, which were compared with the original real data time series. For this, monthly resolution gaps were selected at random which sometimes created gaps of a few months at a time. An average coefficient of determination and root mean square error equal to 0.86 and 1.06 °C was found, respectively, when comparing the synthetic surface temperatures with the real data. Further, we found that the methodology worked best for periods when real temperatures were not extreme. Considering these statistics, on the whole we are confident that the gap-filled temperatures are adequate for estimating trends, but we must keep in mind the potential uncertainty in trend estimates when gaps are large.

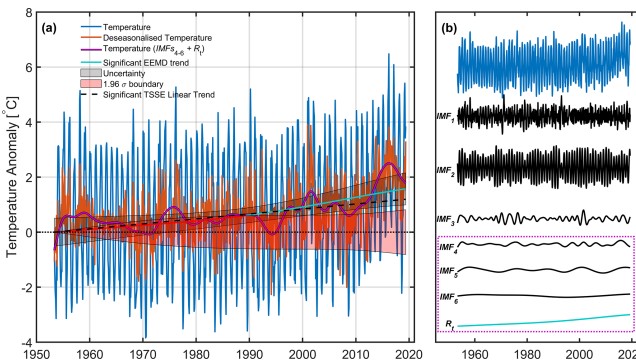

**Figure A1. (a)** Port Hacking monthly temperatures at a depth of 2 m before (blue) and after (purple) $IMF_{1-3}$ (intrinsic mode function) have been subtracted are shown alongside the deseasonalised temperatures for reference (orange) and the trends ($R_t$) estimated using the ensemble empirical mode decomposition (EEMD) method (light blue) and using the Theil–Sen slope estimator (TSSE) method (thick dashed black line). The insignificant CE5 portion of the EEMD trend (red) is also shown. The EEMD trend uncertainty (black patch) estimated using the downsampling method is also displayed, alongside the 95 % confidence range for the null hypothesis that the EEMD trend has arisen by chance from zero-mean stochastic processes (red patch). **(b)** The same monthly temperature data as in **(a)** separated into $IMF_{1-6}$, alongside the same EEMD $R_t$ that is shown in **(a)**. The IMFs used for the purple line in **(a)** is surrounded by a dotted box of the same colour.

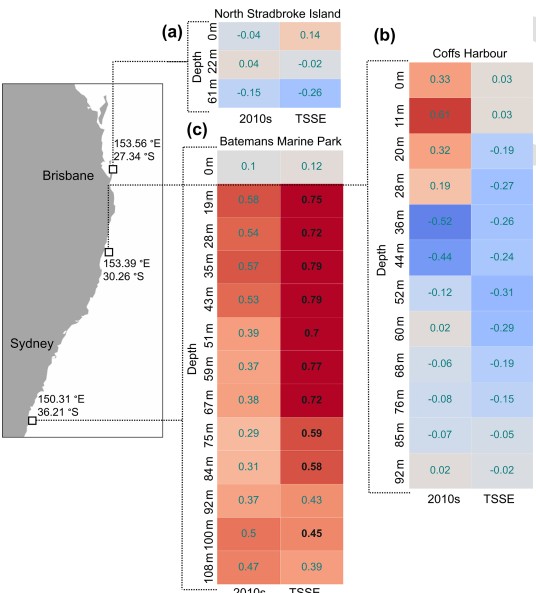

**Figure A2.** Statistically significant (bold black text) and insignificant (light-grey text) average EEMD trend rates and Theil–Sen slope estimator (TSSE) trend estimates (°C per decade) between 2012 and 2020 for multiple depths at **(a)** North Stradbroke Island, **(b)** Coffs Harbour, and **(c)** Batemans Marine Park. The locations of the sites are shown in the left panel, and the approximate depths are shown. Note that all of the EEMD trends shown in this figure have an estimated uncertainty of between 1.2 and 2.3 °C per decade CE6 dependent on site location and depth.

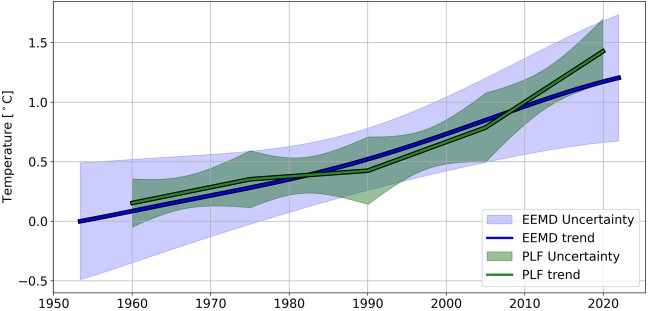

**Figure A3.** The ensemble empirical mode decomposition (EEMD) and piecewise linear fit (PLF) trends and their uncertainty at 2 m depth at the Port Hacking site. For visualisation purposes, an offset of approximately 0.2 °C has been added to the piecewise linear fit trend so as to not start at 0 in 1960. This offset was calculated using the mean difference between the EEMD and piecewise linear fit trends between 1960 and 2020 prior to applying the offset.

## Appendix B: The emd method

The emd CE7 method decomposes a given time series $x(t)$ into a set of oscillatory functions called intrinsic mode functions (IMFs) through a sifting process that

1. connects two cubic splines – one spline through all local minimum points and one spline through all local maximum points in $x(t)$, referred to as the "lower" and "upper" envelopes, respectively;

2. calculates the difference between the mean of the lower and upper envelopes and $x(t)$, producing a new time series $h(t)$;

3. repeats steps 1 and 2 above using $h(t)$ until the lower and upper envelopes are symmetric with a zero mean, with the time series $h(t)$ then being considered an IMF;

4. subtracts the IMF from $x(t)$ to produce a new residual time series $R(t)$ and then repeats steps 1 to 3 using $R(t)$.

This sifting process continues until either $R(t)$ is monotonic or $R(t)$ contains only one extremum. The resulting IMFs and trend are then obtained, separating various modes of variability. We use the MathWorks MATLAB official emd (empirical mode decomposition) function (https://au.mathworks.com/ help/signal/ref/emd.html, last access: 9 November 2022), as used by Stallone et al. (2020), with an input sift relative tolerance (stopping criteria) of 0.4 and default settings for all remaining input parameters.

To highlight how the temperature trends have evolved over time at the long-term sites and to allow for temporal contextualisation of other shorter studies, we show the EEMD trends for each decade on record. We take the mean of the first-order temporal monthly derivative of $R(t)$ for each decade multiplied by 120 to reveal the mean decadal trends.

The EEMD method follows steps 1 to 4 listed above; however the difference is that it is applied to a number of $x(t) +$ white Gaussian noise realisations (forming an ensemble). Multiple IMFs are produced – one set of IMFs for each $x(t) +$ white Gaussian noise realisation time series – and then the average is calculated over all ensemble IMFs. The advantage of the EEMD method is that it reduces mixing between IMFs. In this study, $10\,000$ $x(t) +$ white Gaussian noise realisations were used to obtain each monotonic trend, with the white Gaussian noise having a variance of 0.2 relative to the variance of $x(t)$, as used by Chen et al. (2017).

We apply the EEMD method to each time series from the five sites at each depth. An example from Port Hacking at the surface is shown in Fig. A1. To ensure that IMFs are comparable over depth at a particular site the maximum number of IMFs prior to estimating $R(t)$ were limited, a maximum of six IMFs were chosen for each depth and site. These limits were chosen to derive meaningful $R(t)$ that were either monotonic or near-monotonic functions or contain one extremum. The EEMD method, as with any local analysis method, is affected by edge effects (e.g. "cone of influence" for wavelet analysis) (Torrence and Compo, 1998; Wu et al., 2011). Further, as well as demonstrating this point, Stallone et al. (2020) show the consequence of using the EEMD algorithm for time series containing spikes or jumps. Hence, we use monthly binned time series at all sites to limit the effect of spikes, and we estimate the uncertainty (described below) to better understand the potential influence of edge effects on trend estimates.

The EEMD trends are considered significant if a null hypothesis that the trends have arisen by chance from zero-mean stochastic processes is rejected. The approach used by Ji et al. (2014) and Chen et al. (2017) was used to determine significance, which is briefly summarised below.

1. Compute the lag-1 autocorrelation ($\alpha$) of $x(t)$. If $\alpha = 0$ then the null hypothesis using white Gaussian noise is chosen, while if $\alpha > 0$, as we might expect for ocean time series, then the null hypothesis using red noise is chosen. In our case, lag-1 corresponds to 1 month.

2. Generate 1000 time series of red noise with the same length and standard deviation as $x(t)$. We use 1000 time series here to reduce computation time.

3. Estimate $R(t)$ for each generated time series of red noise using the EEMD method. These 1000 $R(t)$ form an empirical probability distribution function, which at any point in time is approximately normally distributed.

4. Compare the estimated $R(t)$ using $x(t)$ with the $1.96 \times$ standard deviation spread (approximately equal to the 95 % confidence interval) of the generated 1000 red noise $R(t)$. We do not standardise $R(t)$ prior to comparison as the time series of red noise were produced using the standard deviation of $x(t)$.

If the estimated trend is outside of this 95 % confidence interval, then the null hypothesis that the trends are from noise is rejected and those portions of $R(t)$ are considered to be significant (see Fig. A1a). As $\alpha > 0$ at all sites using data after 2010, we generated time series of red noise with similar characteristics to the real data using the Python function "Signalz" (https://matousc89.github.io/signalz/, last access: 12 November 2021).

An uncertainty estimate of the trends was also provided using the downsampling method (Chen et al., 2017; Wu et al., 2011; Wdowinski et al., 2016). For each temperature time series, a monthly temperature is randomly picked for each calendar year, forming a new time series. This is repeated 1000 times, and the trend is estimated for each time series. The mean and standard deviation is then calculated over these 1000 estimates, the latter of which is used as the uncertainty estimate. For the decadal trend estimates (°C per decade, Fig. 4), we provide uncertainties that are calculated by taking the standard deviation of the 1000 subsampled mean rates, which are the means of the first-order temporal derivative of each trend estimate for each decade multiplied by 10. Again, 1000 time series are used here to reduce computation time.

## Appendix C: Acceleration uncertainty

Our results show that the temperature trend is accelerating at Port Hacking and Maria Island when estimated using the EEMD method, albeit with some uncertainty (Fig. 4, $\pm 0.07$–$0.14$ °C per decade). We gain insight into uncertainty that is related to the EEMD methodology rather than from the temperature time series itself by comparing the EEMD trend with one estimated using a piecewise linear fit. Here we show the 2 m depth trend at Port Hacking following the methodology for ocean heat content described by Cheng et al. (2022). Piecewise linear fit segments of 15 years were determined as optimal for deriving a non-linear trend (Cheng et al., 2022); therefore we used four 15-year segments here: 1960–1975, 1975–1990, 1990–2005, and 2005–2020. As with the TSSE trend at this depth, deseasonalised temperatures were used to estimate the piecewise linear fit trend. The downsampling method that was used for estimating uncertainty for the EEMD trends was also used to estimate the piecewise linear fit trend uncertainty.

*Code availability.* Code used for analysis is contained within a Zenodo repository (...) and is available at the following DOI:... TS5

*Data availability.* We use the aggregated temperature data products created by Roughan et al. (2022a) available at https://doi.org/10.26198/5cd1167734d90 (Roughan et al., 2022b). The data sets contained in these aggregated temperature data products are available as follows: historical bottle and CTD profiles from https://www.cmar.csiro.au/data/trawler/regions.cfm; IMOS mooring instrument files,

long time series products, and CTD profiles from https://thredds. aodn.org.au/thredds/catalog/IMOS/ANMN/catalog.html (last access: 9 November 2022) or from https://portal.aodn.org.au/ search (last access: 9 November 2022); and IMOS multi-sensor L3S SST data from http://thredds.aodn.org.au/thredds/ catalog/IMOS/SRS/SST/ghrsst/L3S-1d/ngt/catalog.html (last access: 9 November 2022) or from https://portal.aodn.org.au/ search (last access: 9 November 2022). The SST data trends shown in Fig. 1 and published by Wijffels et al. (2018) are available from https://thredds.aodn.org.au/thredds/catalog/CSIRO/ Climatology/SSTAARS/2017/catalog.html or from https://portal. aodn.org.au/search (search for "SSTAARS").

*Author contributions.* MPH: conceptualisation, methodology, data analysis and investigation, visualisations, writing of original draft. MR: conceptualisation, methodology, writing – review and editing. NM, AS: methodology, writing – review and editing

*Competing interests.* The contact author has declared that none of the authors has any competing interests.

*Acknowledgements.* We thank the three anonymous reviewers that provided suggestions that improved this study. We acknowledge everyone who has been involved in hydrographic sampling since the 1940s and the foresight of CSIRO Marine Research for instigating and continuing the data collection. We acknowledge the NSW-IMOS (New South Wales) moorings and NRS teams for the ongoing mooring deployments and boat-based data collection including Tim Austin, Stuart Milburn, Tim Ingleton, and Clive Holden. At Maria Island the sampling is led by David Hughes and the CSIRO IMOS team. Data were sourced from Australia's Integrated Marine Observing System (IMOS) – IMOS is enabled by the National Collaborative Research Infrastructure Strategy (NCRIS). It is operated by a consortium of institutions as an unincorporated joint venture, with the University of Tasmania as lead agent. This research includes computations using the computational cluster Katana supported by Research Technology Services at the University of New South Wales (UNSW Sydney) (Katana, 2010).

*Review statement.* This paper was edited by Katsuro Katsumata and Mario Hoppema and reviewed by three anonymous referees.

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

## Remarks from the language copy-editor

## Remarks from the typesetter