# Peer review of "Observed multi-decadal trends in subsurface temperature adjacent to the East Australian Current"

_EGUsphere, 2022_

## Author Comment (AC1)

We thank the anonymous referee for reviewing our work.

Can the authors explain why the two particular sites are chosen? Are they selected because of their measurement quality? Because they are the only sites available? Because they are at particularly important locations? What is the reason?

We will add the following text in the manuscript in Section 2.1:

> *"The Port Hacking and Maria Island sites were chosen for two reasons. (1) The sites are long term, containing over 70 years of temperature data and enabling long-term trend detection. (2) One site is close to the EAC separation region, while the other is further downstream in the EAC southern extension region. Hence, we can compare trends at locations with contrasting oceanographic conditions. At Port Hacking there are two long term sites (50m and 100m) but only the 100 m site has moored measurements hence we use data from the 100m site."*

I see a discussion about the accuracy of sensors that are on each of the observing platforms. But what about systematic biases?

It is challenging to explore systematic biases between the observing platforms as we require overlap of measurements in time and space. However, there are instances when this is possible.

For example, Satellite surface temperatures agree well with the nearest subsurface mooring measurements (Roughan *et al*., 2022, Malan *et al*., 2021), keeping in mind that there are sometimes differences related to the exact depth of measurements, stratification, and vertical mixing. Further, the satellite surface temperatures agree well with those retrieved by the surface buoys that have been temporarily in place over time at Port Hacking and Maria Island (Figure 1).

[Figure]

Figure 1: Surface temperature measured at the Port Hacking and Maria Island mooring sites compared with surface satellite temperatures. A 1:1 line is also present coloured blue.

We can also compare CTD profiles (typically collected monthly) with mooring measurements close in time and space at the Port Hacking and Maria Island sites, shown in Figure 2 below.

[Figure]

Figure 2: Comparing mooring temperature measurements at Port Hacking and Maria Island within 10 minutes of CTD profiles collected nearby between 2009 and 2022. A 1:1 line is also present.

There is good agreement between mooring measurements and the CTD profiles, with a mean bias of 0.01 (negative at Maria Island).

We have also compared the bottle samples with CTD profiles at the Port Hacking site when there is spatial and temporal overlap (Figure 3).

[Figure]

Figure 3: Comparing bottle samples with CTD profiles when available at a similar time (< 12 hours) and location (typically < 5 km) between 1989 and 2004.

These comparisons do not indicate any major biases between the different data sets. We will include the following text in Section 2.1 relating to the referee's comment:

> *"Where possible, temperatures observed using the various data platforms close in time and space to one another were matched and compared, and our analysis did not indicate any major systematic biases."*

I don't see a reference or a discussion of figure 3 prior to the appearance of the figure. Perhaps I missed it?

We reference Figure 3 alongside Figure 4 on L155, Section 3.1. It is here where we also discuss the results shown in Figure 3.

In the second paragraph in Section 3.1, what I think are surprising results are discussed. That the warming rates at deeper layers exceed those at shallower layers. This behaviour is discussed in depth in Section 4.1 with plausible physical mechanisms.

Did you mention any QC that is used on these measurements? If that was done, I missed it.

We did not specifically mention data set QC, however the QC was the same through the water column. Rather, we cited Roughan et al., (2022) several times, and this work includes details on QC. But we will add the following sentences to the manuscript in Section 2.1:

> *"All temperature data used in this study have been quality controlled. The historical bottle data were initially quality controlled by the Commonwealth Scientific and Industrial Research Organisation (CSIRO) with evolving practice over time, while IMOS CTD and mooring data collected since 2008/2009 were quality controlled using standardised IMOS procedures (Ingleton et al., 2014; Lara-Lopez et al. 2017; AODN, 2023). For satellite data, quality level flags >= 4 were used to select temperature data. Additional QC checks were performed for all data sets, as described by Hemming et al., (2020), and further information on data quality control is described by Roughan et al., (2022)."*

(Ingleton *et al.*, 2014): https://s3-ap-southeast-2.amazonaws.com/content.aodn.org.au/Documents/IMOS/Facilities/national_mooring/BGCManual_supplementary_documentation/IMOS_NRS_BGC%20Standardised%20Profiling%20CTD%20Data%20Processing%20Procedures_v2.0_Mar2014.pdf

(AODN, 2023): https://github.com/aodn/imos-toolbox/wiki/PPRoutines

(Lara-Lopez *et al.*, 2017): https://repository.oceanbestpractices.org/handle/11329/568

I also wonder how valuable it is to compare against quite old studies (Bindoff 1997 for example).

We had included Holbrook & Bindoff (1997) as it is the only inter-decadal temperature trend off southeastern Australia, to the best of our knowledge, that uses a long-term data set (1955 – 1988) from below the surface. We detect a temperature trend between 1953 and 2022 that is very similar to the one described by Holbrook & Bindoff (1997), suggesting that temperature has increased steadily at depth. We therefore believe this reference is valuable to our discussion.

We also think that it is valuable to compare historic trends over a range of time periods to demonstrate the effect of time period length on temperature trends. We discuss this in Section 4.3.

I would think that limiting to studies done in the past decade or so would be wise. Perhaps they are not available?

To our knowledge we have included all recent studies at the time of submission.

---

## Author Comment (AC2)

We thank the anonymous referee for reviewing our work.

**Major**

The inference that warming is accelerating seems to be (?) based on the magnitude of decadal trends, given in Figure 4, getting larger when computed for more recent decades. However, I struggle to see what statistical evidence there is for acceleration, since you've not explicitly tested for a change in the rate of warming.

The acceleration we identify is based on both the magnitude of the decadal trends in Figure 4, as well as the trends shown in Figure 3. We acknowledge that the trend uncertainty reduces the robustness of the accelerating trends.

We suspected that the higher uncertainty close to the start and end of the time series is dominated by the EEMD methodology. We explored this through estimating the trend using the piecewise linear fit method with 4 x 15-year segments, following the recommendations suggested by Cheng et al. (2022).

The trends and uncertainties using the EEMD and piecewise linear fit methods are compared with one another below in Figure 1. As per our paper, we use the deseasonalised temperatures to estimate the piecewise linear fit trend, and the downsampling method to estimate uncertainty. The comparison in Figure 1 highlights that the higher uncertainty arises from the method rather than from the *in situ* temperature timeseries. Indeed, we see that the trend is also accelerating using the piecewise linear fit method and the EEMD trend is within the uncertainty bounds of the piecewise linear fit trend. Hence, even when keeping in mind the uncertainty, particularly close to the start and end of the timeseries, we are confident that temperatures are accelerating.

The piecewise linear fit uncertainty is higher close to the segment break points, hence, like the EEMD method, the piecewise linear fit method is also sensitive to boundaries.

[Figure]

**Figure 1**: The Ensemble Empirical Mode Decomposition (EEMD) and Piecewise Linear Fit (PLF) trends and their uncertainty at 2m depth at the Port Hacking site. For visualisation purposes, an offset of approximately 0.2°C has been added to the piecewise linear fit trend so to not start at zero in 1960. This offset was calculated using the mean difference between the EEMD and piecewise linear fit trends between 1960 and 2020 prior to applying the offset.

We now include Figure 1 above in the appendix, along with a new appendix section called 'Acceleration Uncertainty'. Here we describe the piecewise linear fit methodology as follows:

> *Our results show that the temperature trend is accelerating at Port Hacking and Maria Island when estimated using the EEMD method, albeit with some uncertainty (Fig. 4, ± 0.07 - 0.14°C*

*decade). We gain insight into uncertainty that is related to the EEMD methodology rather than from the temperature time series itself by comparing the EEMD trend with one estimated using a piecewise linear fit. Here we show the 2 m depth trend at Port Hacking following the methodology for ocean heat content described by Cheng at al., (2022). Piecewise linear fit segments of 15 years were determined as optimal for deriving a non-linear trend (Cheng at al., 2022), therefore we used 4 x 15-year segments here: 1960-1975, 1975-1990, 1990-2005, and 2005-2020. As with the TSSE trend at this depth, deseasonalised temperatures were used to estimate the piecewise linear fit trend. The downsampling method that was used for estimating uncertainty for the EEMD trends was also used to estimate the piecewise linear fit trend uncertainty.*

We have edited text, and added a new paragraph relating to the EEMD uncertainty in Section 4.2 as follows:

*The EEMD method is useful as it shows rates of warming over time, but the uncertainty associated with the EEMD trends has to be considered. We suspect that a large portion of the higher uncertainty that we estimate close to the time series start and end points (Fig 3) is due to the methodology, rather than due to the temperature time series. This is because the EEMD method suffers from edge effects (Stallone et al., 2020). We explored the extension algorithm provided by Stallone et al., (2020) for reducing edge effects, but sensitivity tests indicated that in our case it was better to use the original non-extended time series.*

*To test whether the estimated EEMD uncertainty was predominantly a result of the methodology, we compared the EEMD trend and its uncertainty with a Piecewise Linear Fit trend and its uncertainty (Figure A3), following the methodology described in Appendix C. The piecewise linear fit results also confirms that the warming trend is accelerating over time and has lower uncertainty than the EEMD trend uncertainty. Further, the EEMD trend is within the uncertainty range of the piecewise linear fit trend. This comparison suggests that, despite the higher uncertainty, the EEMD method is useful for identifying a meaningful accelerating trend.*

At the very least, uncertainties are needed on the EEMD decadal trends shown in Figure 4, …

We have added the uncertainties to Figure 4, and the revised figure is below. These uncertainties were calculated as follows (and added to Appendix B):

*For the decadal trend estimates (°C decade, Figure 4), we provide uncertainties that are calculated by taking the standard deviation of the 1,000 subsampled mean rates; that being the means of the first order temporal derivative of each trend estimate for each decade multiplied by 10.*

The uncertainty shown in Figure 3 represents variability in the trend start/end points (offset), while the uncertainty shown in Figure 4 represents variability in the slope of the trend.

We now include this updated figure in the edited version of the manuscript, along with an updated caption (also copied below).

[Figure]

**Revised Figure 4**: Temperature trends for multiple depths at (a) Port Hacking (NRSPHB) and (b) Maria Island (NRSMAI). Statistically significant (black bold text) and statistically insignificant (grey text) average EEMD trend rates per decade (°C decade), and the total time-average over all decades (`Ave.', black text) are shown. The uncertainty for each decade is listed beneath the average EEMD trend rates as white text. The statistically significant Theil-Sen Slope Estimator (TSSE) trend estimates (°C decade) using data over the whole time period are also shown for each site. The locations of the sites are shown in the left panel. The total time-averaged EEMD estimates use both statistically significant and insignificant trend rates over the whole time period, and thus are taken as insignificant estimates. Decade trends are considered significant if 75% or more of the trend during the selected decade are outside the 95% confidence bounds.

(continued) … and some evaluation of these is needed so that the reader can understand whether the trend magnitude during one period (e.g. 2010s) is statistically distinguishable from that during another (e.g. 1950s). After all, you go on to state that there is "high uncertainty" that the warming rate at 2 m at Port Hacking and Maria Island has accelerated over time (L215), but it's not particularly clear to the reader how large these uncertainties are, or how you've reached that conclusion. I note that an uncertainty as large as 0.5 °C decade-1 is given later (L263) for the EMMD trends.

Where applicable we edited the text so that the uncertainties are listed rather than simply stating "high uncertainty", as we agree with the reviewer that this wasn't clear. For example:

> L240 (updated manuscript) *The Port Hacking and Maria Island EEMD warming rates at 2 m depth have accelerated over time (0.2 to 0.25 °C decade⁻¹ during the 2010s), with uncertainty of ±0.11 to ±0.14 °C decade⁻¹ (Fig. 4).*

*We have also edited text elsewhere relating to the EEMD trend uncertainties. These changes are visible in the tracked changes version of the manuscript.*

We discussed the EEMD trend uncertainty in the results section on L171:

> *The uncertainty is approximately 0.5°C decade close to the time period edges (1953 and 2022), and approximately 0.25°C decade between the 1980s and 2000s.*

But we acknowledge that this paragraph is short, so we have edited it as follows:

> *The uncertainty for EEMD trends at both Port Hacking and Maria Island at any given time is shown in Fig. 4. The uncertainty at the surface is approximately 0.5 ◦ C close to the time*

*period edges (1953 and 2022), and approximately 0.3 ◦ C between the 1980s and 2000s. When considering the whole time series, the uncertainty for the decadal trends is shown in Fig. 4, and is approximately ±0.07 to 0.14 ◦ C decade−1. In general, uncertainty below the surface is lower than at the surface, and uncertainty decreases with depth, with the most robust results at the bottom. We tested the uncertainty estimates (as discussed and shown in Appendix C and Fig. A3, respectively) and we are confident that the accelerating rates of warming presented are robust.*

Can acceleration of warming between decades actually be distinguished from these in-situ T timeseries, given such large uncertainties in the warming trends themselves?

As described above, we acknowledge that the trend uncertainty reduces the robustness of the accelerating trends. Figure 1 above highlights that the higher uncertainty close to the time series start and end points arises from the method rather than from the *in situ* temperature timeseries.

We also see that the piecewise linear fit trend is also accelerating and that the EEMD trend is within the uncertainty bounds of the piecewise linear fit trend. Hence, even when keeping in mind the uncertainty, we are confident that temperatures are accelerating.

**Minor**

2. L145-147. I think (?) you mean that the decadal trend is estimated by taking "…the mean of the first order temporal monthly derivative of the EMMD *R(t)* for each decade…", rather than "…the mean of the first order temporal monthly derivative of the EMMD temperature trend for each decade…".

We have modified the text as follows:

*"…the mean of the first order temporal monthly derivative of the EEMD temperature trend (R(t)) for each decade…"*

'R(t)' is mentioned in Appendix B only, hence for clarity we will only include this in brackets here.

3. L152. It's really hard to see the white dashed lines in Figure 3.

We have improved the figure (also copied below with edited caption) so that significant trend periods have a filled colour with a thick black outline and insignificant periods have instead a dashed line. We think the significant/insignificant periods are now easier to distinguish.

[Figure]

**Revised Figure 3**: The temperature Ensemble Empirical Mode Decomposition (EEMD) trends at each of the five sites: (a) Port Hacking, (b) Maria Island, (c) North Stradbroke Island, (d) Coffs Harbour, and (e) Batemans Marine Park. Each colored line represents a depth level (metres) at each of the sites, as indicated in the corresponding legends. The uncertainty for each depth level estimated using the downsampling method is represented by the shaded area with the colour corresponding to the lines. Significant trend periods are represented by a filled line with a black outline. Insignificant trend periods are indicated by dashed lines. Note the difference in y- and x-axis limits between panels (a-b) and (c-e).

4. L160-161. Can you please clarify how you are "detecting" acceleration here? Presumably based on the trend magnitude increasing between decades? This sentence also implies that the acceleration is not statistically significant until the 1990s, but it's unclear in the manuscript what method you are using to assess this?

We have edited this sentence as follows:

> *At most depths EEMD trend acceleration is detected from the 1970s, although not statistically significant until the1990s as determined by the methodology described in Appendix B.*

5. L247. I'm not sure that Shears and Bowen (2017) "emphasize acceleration". The decadal trends at Maria Island they present during 1982-2016 are larger than between 1946-2016, but the uncertainties on these trends are large and they overlap with one another. Therefore, it's questionable whether any acceleration is present and/or statistically supported.

We have edited this sentence as follows:

> *Shears and Bowen (2017) also present possible acceleration at Maria Island providing temperature trends of 0.2◦ C decade−1 from 1946 to 2016, and 0.32◦ C decade−1 from 1982 to 2016, respectively, keeping in mind that these trends likely have considerable uncertainty and overlap with one another.*

6. L355-364. Based on the text in Appendix B, I was confused as to how you extract a final value (e.g. °C decade-1) for the "monotonic trend" of *x(t)* using EMMD. I think this information is given in the main text on L145-147: it is the mean of the first order temporal monthly derivative of *R(t)*. Please clarify how you end up with trends in °C decade-1 in Appendix B.

We have included the following text on how we extract decadal trends in Appendix B (L384):

> *To highlight how the temperature trends have evolved over time at the long-term sites, and to allow temporal contextualisation for other shorter studies, we show the EEMD trends for each decade on record. We take the mean of the first order temporal monthly derivative of R(t) for each decade multiplied by 120 to reveal the mean decadal trends.*

---

## Author Comment (AC3)

We thank the anonymous referee for reviewing our work.

**General comments:**

Section 2.1 - the paper makes use of a number of different instruments and datasets with varying instrument accuracies. However it is not clear how these different instrument accuracies are accounted for and whether these may or may not impact the overall trend analysis.

The dataset accuracies are high, with nominal accuracies of between ±0.002 and ±0.05°C (Roughan *et al.*, 2022). Due to their small size these accuracies are not explicitly accounted for in our trend analysis, however, they fall within the range of uncertainty.

The acronyms are unnecessarily complicated. The author also uses the acronym and the full name for the sample site interchangeably within the text.

We have edited the figures and text to always use the site names, rather than the acronyms as follows:

> NRSPHB -> Port Hacking
>
> NRSMAI -> Maria Island
>
> NRSNSI -> North Stradbroke Island ('NS Island' in Figure 6 to save space)
>
> CH100 -> Coffs Harbour
>
> BMP120 -> Batemans Marine Park ('Batemans MP' in Figure 6 to save space)

Line 129 - avoid confusion by renaming the combined trend method after one of the trends used.

In addition to the text on L129, we have edited the text after first referring to the TSSE trend on L173 as follows:

> *Trends estimated using both the EEMD and TSSE (combined Mann-Kendall and Theil-Sen Slope Estimator) methods are compared at Port Hacking and Maria Island …*

Line 149 - "expect trends to be sensitive to the time period choice" - do you show this anywhere in the paper by way of explanation?

We have modified the text on L149 as follows:

> *Although data are available since 1944 at Maria Island, we estimate long-term temperature trends at this site between 1953 and 2022 for consistency with Port Hacking. , as we expect trends to be sensitive to the time period choice*

The Discussion would be easier to understand if you follow the same format as your Results section. It reads a little disjointedly. Section 4.2 would probably fit better within the Methods section. There is also a repetition of information through the Discussion section (e.g. line 274-279).

We have reorganised the discussion section to follow a similar format as the results section. We have separated Section 4.1 into 2 subsubsections:

> *4.1 Contextualising the observed trends*
>
> *4.1.1 Surface Warming*
>
> *4.1.2 Subsurface Warming*

We have decided to keep section 4.2 in the discussion section as it relies on the uncertainty estimates that we show in the results section. Additionally, we have added text in Section 4.2 relating to analysis comparing the Port Hacking 2 m accelerating EEMD trend with a piecewise linear fit in response to a comment posted by Reviewer 2. We do not believe this additional text would be better placed in the methods section.

We have also removed repetition in the discussion.

Line 300 - You refer here to "Sydney" which is not noted on your Figure 1, nor previously in text. Please amend as not all your readers will be local.

We have changed "*Sydney*" to "*Port Hacking*" in the text here.

Line 8 - "at in the top 20 m" ?

We have changed this to:

> *...in the top 20 m...*

Line 21 - comma after "(...2019)"

We have added a comma here.

Line 73 - "...starting in the 1940/50s to present"

Line 126 - Finish sentence after (Hamed and Rao, 1998), Start new sentence with "As..."

We have modified the text on L73 and L126 as suggested by the reviewer.

---

## Author Response (AR2)

The authors have made the following changes as suggested by the editor:

L.138. Appendix.Sect.C → Appendix. B

We have modified the text here, and further removed all instances of 'Sect.' after 'Appendix'.

L.380. 'upper' and 'lower' → 'lower' and 'upper' (or swap 'minima' & 'maxima' on L.379?)

We have modified all instances of 'upper and lower' (or similar) to be 'lower and upper' instead in Appendix B for consistency with the text on L.379.

Additional minor changes:

As discussed with the editor, we have now included a code availability statement section. We plan to edit this section to include a DOI and citation during the typesetting stage after we make the code available on Zenodo. We expect to publish the code on Zenodo early next week.